

# Atmospheric Lifetimes, Infrared Absorption Spectra, Radiative Forcings and Global Warming Potentials of $NF_3$ and CFC-115

Anna Totterdill[1], Tamás Kovács[1], Wuhu Feng[1,2], Sandip Dhomse[3], Christopher J. Smith[4], Juan Carlos Gómez–Martín[1], Martyn P. Chipperfield[3], Piers M. Forster[3] and John M. C. Plane[1]

[1] School of Chemistry, University of Leeds, Leeds LS2 9JT, UK

[2] NCAS, School of Earth and Environment, University of Leeds, Leeds LS2 9JT, UK

[3] School of Earth and Environment, University of Leeds, Leeds LS2 9JT, UK

[4] Energy Research Institute, School of Chemical and Process Engineering, University of Leeds, Leeds LS2 9JT, UK

*Correspondence to*: Piers Forster (P.M.Forster@leeds.ac.uk)

**Abstract.** Fluorinated compounds such as $NF_3$ and $C_2F_5Cl$ (CFC-115) are characterised by very large global warming potentials (GWPs) which result from extremely long atmospheric lifetimes and strong infrared absorptions in the atmospheric window. In this study we have experimentally determined the infrared absorption cross-sections of $NF_3$ and CFC-115, calculated the radiative forcing and efficiency using two radiative transfer models and identified the effect of clouds and stratospheric adjustment. The infrared cross sections are in good agreement with previous measurements, whereas the resulting radiative forcings and efficiencies are, on average, around 10% larger. A whole atmosphere chemistry-climate model was used to determine the atmospheric lifetimes of $NF_3$ and CFC-115 to be $(616 \pm 34)$ years and $(492 \pm 22)$ years, respectively. The GWPs for $NF_3$ are estimated to be 14600, 19400 and 21400 over 20, 100 and 500 years, respectively. Similarly, the GWPs for CFC-115 are 6120, 8060 and 8630 over 20, 100 and 500 years, respectively.

## 1 Introduction

Fluorinated compounds such as $NF_3$ and CFC-115 are potentially important for global warming [*Myhre et al.*, 1998]. The stability and thermo-physical properties of these gases have made them attractive chemicals for use in many relatively modern industrial processes. $NF_3$ is being used increasingly as a replacement for banned perfluorocarbons (PFCs) which were utilised in processes such as chemical cleaning and circuit etching. It is an extremely potent greenhouse gas with an estimated 100 year Global Warming Potential (GWP) between 10,800 and 17,000 [*Robson et al.*, 2006; *Weiss et al.*, 2008; *Arnold et al.*, 2013]. *Weiss et al.* [2008] reported a 2008 mean global tropospheric mixing ratio of 0.45 ppt increasing at a rate of 0.053 ppt yr$^{-1}$. The present day mixing ratio will therefore be close to 1 ppt, assuming no change in emission rate.



*Arnold et al.* [2013] found undetectable levels of $NF_3$ prior to 1975 in archived air samples and ice cores, indicating that the major source of the gas is anthropogenic.

CFC-115 was introduced as refrigerant in the 1970s and prior to this was not detected in the atmosphere. Following the phasing out of CFC-115 through the 1997 Montréal Protocol [*Solomon et al.*, 2007], its atmospheric concentration

stabilized by 2005. A mixing ratio of 8.4 ppt was reported in 2011, with a decreasing trend of 0.01 ppt yr$^{-1}$ [*Maione et al.*, 2013]. Based on an atmospheric lifetime of 1020 years CFC-115, its 100 year GWP was estimated to be 7370 [*Solomon and Miller*, 2007].

The change in concentration of any trace gas depends in part on how its emission evolves over time, but also on the rates of any chemical and physical removal processes. The only important sinks for CFC-115 and $NF_3$ appear to be

photolysis [*Dillon et al.*, 2010; *Totterdill et al.*, 2014; *Totterdill et al.*, 2015] and reaction with $O(^1D)$ [*Dillon et al.*, 2011; *Baasandorj et al.*, 2013]. Reaction with the meteoric metals Na and K is a minor loss route [*Totterdill et al.*, 2014; *Totterdill et al.*, 2015].

Trace gas concentrations are dependent on the atmospheric lifetimes ($\tau$) of the species, defined as the ratio of the total atmospheric burden to the total loss rate. Estimates of the atmospheric lifetimes of $NF_3$ and CFC-115 have recently

been reported by *SPARC* [2013]. For $NF_3$ the recommended value is 569 years, which was based on 2-D model calculations including loss due to photolysis (71% of total loss) and reaction with $O(^1D)$ (29%). This overall lifetime was slightly larger than the value of 500 yr given in the *WMO* [2011] assessment. For CFC-115, the recommended lifetime is 540 yr (37% of loss due to photolysis, 63% due to reaction with $O(^1D)$), which was much lower than the *WMO* [2011] value of 1020 yr based on previous photolysis cross sections from *Sander et al.* [2011].

Therefore, these fluorinated compounds are very potent global warming agents because, in addition to having atmospheric lifetimes of many centuries, they absorb infrared radiation strongly between 800 and 1200 cm$^{-1}$. This region of the electromagnetic spectrum is known as the 'atmospheric window' because of a pronounced minimum in atmospheric absorption by $H_2O$, $CO_2$ and $O_3$. Furthermore, the window overlaps with the peak in the terrestrial infrared spectrum (500 - 1500 cm$^{-1}$), making it a particularly important region in the radiative balance of the atmosphere [*Pinnock et al.*, 1995].

However, it is difficult to quantify surface temperature changes resulting from small perturbations due to climate variability and large uncertainties in climate feedback mechanisms [*Pinnock et al.*, 1995]. The historic effects of various drivers of climate change are typically specified and compared in terms of their radiative forcings, a measure of the perturbation to the Earth's energy budget. Various flavours of radiative forcing exist [*Myhre et al.*, 1998]. The effective radiative forcing measures top-of-atmosphere energy budget changes following adjustments to the vertical temperature profile, clouds and

land-surface temperatures. The stratospherically adjusted radiative forcing is defined as the change in net (i.e., down minus up) irradiance at the tropopause (solar plus longwave, in Wm$^{-2}$) after allowing for stratospheric temperatures to readjust to radiative equilibrium, but with the surface and tropospheric temperatures and state held fixed at the unperturbed values [*Ramaswamy et al.*, 2001]. Note the instantaneous radiative forcing (IRF) can be obtained by not applying the stratospheric adjustment. Although effective radiative forcing (ERF) estimates are more representative of temperature changes they are



more uncertain as they rely on climate model estimates of cloud response [*Sherwood et al.*, 2015]. Further, climate model radiation codes do not typically represent minor greenhouse gases (GHGs), therefore it is not currently possible to estimate the ERF for the species considered here. We have therefore estimated RF and IRF using the line-by-line radiative transfer model (RTM). As the RTM only accounts for absorption, the extension to clouds and scattering processes was performed by a secondary radiative transfer model using atmospheric optical depth profiles generated by the RTM.

The purpose of this work was to determine new values for the GWPs of NF$_3$ and CFC-115, based on their cloudy sky adjusted radiative efficiencies (the definition of the GWP is discussed in Section 6). In order to achieve this, infrared absorption cross-sections for both NF$_3$ and CFC-115 were measured and then used as input into the Reference Forward Model (RFM) [*Dudhia*, 2014] and Library for Radiative Transfer (libRadtran) [*Mayer and Kylling*, 2005], two radiative transfer models used to calculate radiative forcings and efficiencies. Here, radiative forcing refers to a perturbation of the modern day concentration of the compound against its pre-industrial concentration, and is given in units of Wm$^{-2}$. Radiative efficiency refers to a perturbation of $0 - 1$ ppb and is given in units of Wm$^{-2}$ ppbv$^{-1}$. The sensitivity of these determined forcings to a number of criteria including cloudiness and stratospheric adjustment was also examined. The atmospheric concentrations of NF$_3$ and CFC-115 were then determined using the Whole Atmosphere Community Climate Model [*Garcia et al.*, 2007], incorporating the chemical loss processes described in our recent papers [*Totterdill et al.*, 2014; *Totterdill et al.*, 2015]. It also produced revised estimates of the atmospheric lifetimes of the two compounds.

## 2 Experimental

The IR spectrum of NF$_3$ has been measured previously by *Robson et al.* [2006] and *Molina et al.* [1995], and that of CFC-115 by *McDaniel et al.* [1991]. There is some deviation across existing the literature cross-sections, and few quantitative full spectra measurements are available. This work was consequently carried out in order to provide a more complete set of measurements and reduce uncertainty in the published data.

Measurements were made in a 15.9 cm gas cell sealed with KBr windows, which allowed transmission between 400 and 40000 cm$^{-1}$. Spectra were recorded with a Bruker Fourier Transform spectrometer (Model IFS/ 66) fitted with a mid-infrared source and beam-splitting optics. Room temperature ($296 \pm 2$ K) measurements were carried out between 400 and 2000 cm$^{-1}$ at a spectral resolution of 0.1 cm$^{-1}$. Absorption spectra were obtained by averaging 128 scans at a scan rate of 1.6 kHz.

Gas mixtures were made using between 12 - 307 Torr of NF$_3$, and 6 - 77 Torr of CFC-115, made up to 760 torr with N$_2$. Multiple mixtures were made up so that the cross-section could be obtained at a selected wavelength $\lambda$ by taking the slope of the linear regression of the corresponding peak absorbance against concentration according to the Beer–Lambert Law:

$$A(\lambda) = \sigma(\lambda)lc \tag{E1}$$

where $A$ is the absorbance, $\sigma$ is absorption cross-section in cm$^2$, $c$ is concentration in molecule cm$^{-3}$ and $l$ is path length in cm. Concentrations were determined from pressures measured with a capacitance manometer (Baratron model 222 CA),



calibrated with an oil manometer. Analyte concentrations were selected so that $A < 1$, to avoid deviation from the Beer – Lambert linear behaviour. Baseline and background corrections (including removal of $CO_2$ and $H_2O$) were performed after the experiments.

Reactant gas mixtures for the experiments were prepared on all-glass vacuum lines. The gases $N_2$ (99.9999%, BOC), $NF_3$ (99.99%, BOC), were used without further purification. Samples of CFC-115, provided by Professor William Sturges (University of East Anglia), were purified by freeze-thaw-pump degassing on a glass vacuum line.

## 3 Atmospheric Modelling

The atmospheric distributions of $NF_3$ and CFC-115 were simulated using the 3-D Whole Atmosphere Community Climate Model (WACCM) [*Garcia et al.*, 2007]. WACCM is a comprehensive numerical model extending vertically from the ground up to the lower thermosphere (~140 km) and is part of the National Centre for Atmospheric Research (NCAR) Community Earth System Model (CESM) [*Lamarque et al.*, 2012]. WACCM calculates the concentrations of atmospheric species by considering all relevant chemical and dynamical processes. Here we have used a free running version of WACCM 4 [*Garcia et al.*, 2007] which has 66 levels from the surface to $5.96 \times 10^{-6}$ Pa with a vertical resolution of 3.5 km scaled height in the mesosphere and lower thermosphere (MLT) region and $1.9° \times 2.5°$ (latitude $\times$ longitude) horizontal resolution. The model contains all the important details of the MLT processes including radiative transfer, auroral processes, non-local thermodynamic equilibrium and the molecular diffusion of constituents.

A series of $NF_3$ and CFC-115 tracers were included in the model. For the model simulations presented here, three loss processes for $NF_3$ and CFC-115 were included: reactions with $O(^1D)$; reactions with the mesospheric metals (Na, K); and UV photolysis. For each compound, a set of 5 tracers were used. One tracer was removed by all three processes; three tracers were used to determine the individual impact of each loss process acting alone; and the fifth was a passive (i.e. chemically inert) tracer.

The relevant rate constants for $O(^1D)$ and the metal atom reactions that were added to the chemistry module in WACCM are listed in Table 1. Note that we have only included the reactions with Na and K, as the reactions with the more abundant meteoric metals Fe and Mg are very slow at temperatures below 300 K [*Totterdill et al.*, 2014; *Totterdill et al.*, 2015]. The standard chemistry scheme in WACCM contains 59 species and 217 gas-phase reactions [*Kinnison et al.*, 2007]. $O(^1D)$ was determined from this scheme. Modules with a further 61 reactions describing the chemistry of Na and K were then added to this scheme, along with the meteoric input functions required to simulate the Na and K layers in the MLT [*Marsh et al.*, 2013a; *Plane et al.*, 2014].

The photolysis rates of $NF_3$ and CFC-115 were calculated in WACCM using the fitted expressions we determined previously for their absorption cross sections as a function of wavelength between 121.6 nm and 200 nm [*Totterdill et al.*, 2014; *Totterdill et al.*, 2015]. Following the study by *Marsh et al.* [2013b], the daily solar spectral irradiances used in WACCM were specified from the model of *Lean et al.* [2005], updated with the total solar irradiance of *Kopp and Lean*



[2011]. The model was then run from year 2000 for 13 years when the solar data is available and is enough for the atmospheric lifetimes of NF$_3$ and CFC-115 to reach steady-state.

## 4 Radiative Transfer Modelling

Radiative forcing calculations were made using the Reference Forward Model (RFM) [*Dudhia*, 2014]. The RFM is a line-by-line radiative transfer model based on the previous GENLN2 model [*Edwards*, 1987], and includes absorption cross-sections for CFC-115 and NF$_3$ derived from the HITRAN database [*Rothman et al.*, 2013]. In addition to providing upwelling and downwelling longwave fluxes for calculating the clear-sky forcing, the RFM was used to generate optical depth profiles at a resolution of 1 cm$^{-1}$ for input into the DISORT radiative transfer solver as implemented in the libRadtran [*Mayer and Kylling*, 2005]. The clear-sky fluxes obtained from the RFM were validated against results from libRadtran for the cloudless, non-scattering case.

Calculations to obtain the IRF and RF were performed using the flux form of the RFM at a spectral resolution of 0.1 cm$^{-1}$, determined by the resolution of the IR spectra measured in the present study (Section 2). The radiative transfer calculation was performed on each spectral band between 550 – 2000 cm$^{-1}$, and the irradiance flux integrated over wavelength to obtain the net irradiance at each level in the model atmosphere. For these calculations integration over zenith angle was performed via a first moment double-Gauss procedure with 8 streams, where the Planck function was set to vary linearly with optical depth. For the stratospheric adjustment, the stratosphere temperatures were adjusted using an iterative process based on heating rate changes that after 100 days have changed the stratospheric temperatures and returned them to radiative equilibrium (see also in Introduction). For both NF3 and CFC115 the temperature change at the tropopause is ~1.5 × 10$^{-5}$ K for a perturbation of 1 ppt (average tropospheric volume mixing ratio), and ~0.01 K for a perturbation of 1 ppb.

A compilation of line data for background species was obtained from HITRAN 2012 [*Rothman et al.*, 2013] and absorption cross sections for NF$_3$ and CFC-115 were measured in the present study. The temperature, pressure and the mixing ratios of the major atmospheric constituents CO$_2$, H$_2$O, CH$_4$, N$_2$O and O$_3$, as well as NF$_3$ and CFC-115 were obtained from the WACCM output. The temperature dependence of background species' absorption is automatically interpolated from HITRAN data. Although the temperature dependence of the NF$_3$ and CFC-115 cross sections was not measured, we assume that it makes a negligible uncertainty in the radiative forcing calculations.

The effect of seasonal and geographical variations on factors influencing radiative forcing means that multiple averaged local radiative forcings calculated across the location-time grid are needed to estimate global forcing. The instantaneous and adjusted radiative forcings and efficiencies were first calculated in the RFM for each month between -90º and 90º, at a 9º latitude and longitude resolution. We also created vertical profiles by averaging three latitudes (representing the tropics, mid- and high-latitudes) for each month. The area averaged forcing from the three profiles was found to yield a forcing value within 2% of that obtained from the gridded data. Compared to using a global annual mean profile, *Freckleton et al.* [1998] demonstrated that the method of averaging a small number of latitudes is 5-10% more accurate than use of the global annual mean profile. Explanations for this difference in accuracy are discussed in Section 5.3. The profiles



representing the three latitudes for each month were also used to calculate radiative forcings and efficiencies in libRadtran. Clear-sky IRFs calculated in libRadtran were within 3% of those calculated by the RFM.

## 5 Results

### 5.1 Infrared Absorption Spectra

Figure 1 illustrates the IR absorption cross section spectrum measured for $NF_3$ in the present study. The band-integrated cross sections obtained from this measured spectrum are listed in Table 2, which also compares with the integrated cross sections reported in the literature, where available. Similarly, the CFC-115 spectrum is displayed in Figure 2, and Table 3 lists the corresponding integrated cross sections and comparison with the literature. The uncertainties in the sample concentrations of $NF_3$ and CFC-115 were ±0.8 and 0.7%, respectively. The average spectral noise was ±5 × $10^{-21}$ $cm^2$ molecule$^{-1}$ per 1 cm$^{-1}$ band. However, at wavenumbers < 550 cm$^{-1}$, towards the edge of the mid IR where the opacity of the beam-splitting filter increased, this increased to ±1 × $10^{-20}$ $cm^2$ molecule$^{-1}$ per 1 cm$^{-1}$ band. The error in scaling the cross sections for $NF_3$ and CFC-115 were estimated to be ±5 and 6%, respectively (error in pressure dependence). This results in an average overall spectral error of ±6 % (1σ) in both cases. This is the average error of the integrated band cross section and is fairly uniform for the major bands.

As shown in Table 2, the intensities of the two main absorptions bands (840-960 and 970-1085 cm$^{-1}$) of $NF_3$ measured in the present work are 3% and 11% larger than those reported by *Robson et al.* [2006], with an average deviation of 29% over the minor bands and 4% across the entire spectrum. All differences excluding the minor bands in the region between 1085 and 1580 cm$^{-1}$ are comfortably within the combined error of both experiments. In contrast, the intensities of the two main absorption bands are 79% and 39% larger than those reported by *Molina et al.* [1995]. The minor bands in our spectrum are on average 32% larger, and the absorption intensity is 42% larger across the whole spectrum. These differences are greater than the combined error from both experiments. Explanations for the lower values reported by *Molina et al.* [1995] have been discussed by *Robson et al.* [2006].

Figure 3 shows that the intensity of the main CFC-115 absorption band (1212-1265 cm$^{-1}$) measured in the present work is 6% smaller than that reported by *McDaniel et al.* [1991]. The other significant bands at 946-1020, 1105-1150, 1160-1212 and 1326-1368 cm$^{-1}$ are 9%, 6% and 3% smaller, respectively. These results are well within the combined error of both experiments.

### 5.2 Atmospheric lifetimes

The monthly-averaged $NF_3$ and CFC-115 concentrations and loss rates in each WACCM grid box were used to estimate the atmospheric lifetimes of $NF_3$ and CFC-115, which were computed by dividing the global atmospheric burden of each compound by its integrated loss rate. The total loss rates were obtained from the sum of the individual loss rates due to photolysis, and reactions with mesospheric metals and with $O(^1D)$.



Figure 3a,b show the globally averaged profiles of the different modelled $NF_3$ and CFC-115 tracers, which illustrate the impact of the different loss processes. The largest mixing ratio profiles for both $NF_3$ and CFC-115 are shown by the passive tracers as they are not subject to the removal processes. Note that the decay of the passive tracer mixing ratios above about 85 km is due to the very long timescale for the tracers to mix vertically into this region. Clearly, the reactions with atmospheric metals do not contribute to the atmospheric removal of these gases (i.e. the metal loss tracers profiles are almost identical to the passive profiles) and hence their impact on the lifetimes is negligible. The reason is that their temperature dependent rate constants are much smaller than the rate constants of $O(^1D)$ reactions and also, as it is shown in Fig. 11 in [*Totterdill et al.*, 2014] ($NF_3$) and in Fig. 9a in [*Totterdill et al.*, 2015] (CFC-115), the removal rates by mesospheric metals are two and four orders of magnitudes smaller for $NF_3$ and CFC-115, respectively than the removal rates by VUV photolysis even at the peak of mesospheric metal layers (90 km). In contrast, photolysis and the reactions with atmospheric $O(^1D)$ are the dominant removal processes. Figure 3c,d are the corresponding monthly averaged zonal mean profiles of $NF_3$ and CFC-115 from the surface up to 140 km in January. The surface mixing ratios of $NF_3$ and CFC-115 uses 1.05 and 7.9 pptv, respectively. Both tracers are well mixed in the troposphere below 15 km. There are sharp decreases in the tropical tropopause layer (TTL) (20–28 km). The mixing ratios of $NF_3$ and CFC-115 are decreasing with increasing altitude from lower stratosphere to the mesosphere/lower thermosphere up to 90 km. Above 90 km, their ratios are quite small ($< 0.1$ ppt for both compounds). Around 40-50 km in the Southern polar region, the low values of $NF_3$ and CFC-115 are caused by the removal of the reactions with $O(^1D)$ (see next). Figure 4 shows the annual mean contributions of photolysis and reaction with $O(^1D)$ to the loss of $NF_3$ and CFC-115. The region of greatest loss is in the tropics, while at high latitudes the removal rates are orders of magnitude smaller. For both $NF_3$ and CFC-115 the dominant region of photolysis is in the stratosphere, below 50 km, although CFC-115 shows a weak secondary peak in photolytic loss over $60 - 80$ km, which is due to the increased photolysis loss rate of 2 orders of magnitude over this range (see Fig. 9b in [*Totterdill et al.*, 2015]). Figure 5 is the annually averaged atmospheric lifetime for $NF_3$ and CFC-115 as a function of simulation time. The length of the model run is sufficient for the tracers to mix below 85 km where the dominant loss processes occur shown in Figure 4. The mean steady-state lifetime of $NF_3$ is then $(616 \pm 34)$ yr, which is 8% larger than the recently published value of 569 yr [*SPARC*, 2013]. Steady-state was assumed to be reached when the lifetime did not change more than 1% between two consecutive years. The uncertainties were estimated from the error bars of the removal rate constants only (see Table 1) using the error propagation. Photolysis and reaction with $O(^1D)$ account for 60.0% and 40.0% of the global removal rate, respectively. The corresponding percentages from *SPARC* [2013] are 71.3% and 28.7%. In the case of CFC-115, the lifetime in the present study is $(492 \pm 22)$ yr, which is 9% smaller than the value of 540 yr from *SPARC* [2013]. Photolysis and reaction with $O(^1D)$ account for 34.4% and 65.6% of the global removal rate, respectively. Again, these are close to the corresponding percentages from *SPARC* [2013], which are 37.4% and 62.6%. The differences between our results and those of *SPARC* [2013] likely arise from differences in the rates of atmospheric circulation in the 3-D WACCM and the 2-D model used in the SPARC report, as well as the implementation in WACCM of recently updated photolysis parameters [*Totterdill et al.*, 2014; *Totterdill et al.*, 2015].



### 5.3 Calculation of Radiative Forcings and Efficiencies

The calculation of radiative forcing and radiative efficiency are sensitive to several factors such as the choice of tropopause height and the way cloud is included in the model. This section examines these sensitivities and further examines the seasonal and latitudinal variations in forcing.

### 5.3.1 Tropopause

The definition of tropopause height directly influences the calculation of radiative forcing. There are three commonly used definitions: the thermal tropopause (ThT), defined as the lowest level at which the temperature lapse rate between this and all higher levels within 2 km falls below 2 K km$^{-1}$ [*WMO*, 2007]; the temperature minimum tropopause (TMT), the base of the stratospheric temperature inversion; and a uniform pressure level of 200 hPa as a proxy for the top of the convective level where there is a significant change in stability below the thermal tropopause [*Forster and Shine*, 1997]. *Forster and Shine* [1997] identified the latter as the most appropriate for radiative forcing calculations at high horizontal resolution. However, the thermal tropopause (ThT) used in this study was found to generate results which were accurate to within 0.5% of those produced by the convective tropopause [*Freckleton et al.*, 1998]. At lower horizontal resolution this uncertainty is greater.

The temperature profile and thus tropopause height show a significant spatial variation and are affected by profile averaging. This adds extra inaccuracies when using a global annual mean profile which does not account for the variation in tropopause height. This effect was explored with respect to the instantaneous radiative forcing of NF$_3$. Figure 7 shows the latitudinal variation of instantaneous radiative forcing with tropopause height when using the ThT, TMT and globally averaged tropopause definitions. The globally averaged thermal tropopause was found to be 12.8 km. When the spatially varying thermal tropopause was applied to the forcing calculations it yielded an average instantaneous radiative forcing 10% lower than that employing the ThT. The globally averaged TMT was found to be 14.9 km. When this was applied it resulted in an average instantaneous radiative forcing 5% higher than the ThT. The spatially averaged TMT was found to be 3% higher on average, compared to its global averaged profile.

The global mean TMT gives a significant overestimation of radiative forcing from a high, unrepresentative tropopause of 14.9 km caused by temperature variations being smoothed out through averaging. Additionally, the averaging procedure affects parameters such as the H$_2$O vapor and O$_3$ profiles, so that averaging monthly profiles over latitudes representing the topics and mid-latitudes (rather than globally) gives a better representation of these variables. Because both NF$_3$ and CFC-115 are well mixed in the stratosphere they are potentially less affected by tropopause height than species which decay strongly in the lower stratosphere [*Myhre et al.*, 1998]. Consequently, the spatially averaged ThT was selected for the calculations described below.



### 5.3.2 Seasonal - Latitudinal Variation

Figure 7 shows the large seasonal-latitudinal impact on variation in the IRF and RF of $NF_3$ and CFC-115 under clear sky conditions. The latitudinal variation in zonally-averaged forcing in a single month can be as large as a factor of 8; the variation in monthly forcing for a single latitude is much smaller, approximately a factor of 2 on average. The lack of uniformity across this grid demonstrates the requirement for higher resolution calculations.

The variation of radiative forcing and efficiency as a function of latitude is primarily due to changes in the Planck function. Differences in cloudiness and $H_2O$ density levels were also found to contribute. Forcings averaged  across the Southern Hemisphere were approximately 25% lower than those averaged across the Northern Hemisphere [*Prather and Hsu*, 2008]. The lowest radiative forcings for each month are observed at the South Pole with the very lowest occurring at the winter Antarctic polar vortex.

### 5.3.3 Cloudiness

Because clouds absorb across the same spectral region as $NF_3$ and CFC-115, their presence will cause a reduction in radiative forcing. Consideration of cloud coverage is therefore crucial to forcing calculations. The treatment of clouds involves determination of cloud band transmittance from user specified liquid water path, effective radius and cloud fraction at each altitude level. The zonal mean coverage for a given latitudinal band obtained as monthly means from the International Satellite Cloud Climatology Project (ISCCP) D2 dataset averaged from between 1983 and 2008 [*Rossow et al.*, 1996]. Results were calculated from different, weighted combinations of clear sky plus various configurations of cloud coverage using the independent pixel approximation in libRadtran.

The IRF and RF in clear and cloudy sky conditions for $NF_3$ and CFC-115 are given in Table 4, and the relative radiative efficiencies in Table 5. Radiative forcings are shown in Figure 6. The globally averaged instantaneous radiative forcings of $NF_3$ and CFC-115 are $2.8 \times 10^{-4}\,Wm^{-2}$ and $19 \times 10^{-4}\,Wm^{-2}$ respectively. The instantaneous radiative efficiency of $NF_3$ is $0.35\,Wm^{-2}\,ppb^{-1}$ which is slightly larger than the different models estimations ($0.248 – 0.252\,Wm^{-2}\,ppb^{-1}$) by [*Robson et al.*, 2006]. The calculated IRE under the cloudy condition for CFC-115 ($0.20\,Wm^{-2}\,ppb^{-1}$) is about 10%  ($0.02\,Wm^{-2}\,ppb^{-1}$) larger than Myhre et al. [2013]. These tables indicate that radiative forcings and efficiencies increase by $30 – 40\%$ where clouds are neglected.  This is because a cloud acts a blackbody and so the contribution to the total forcing from greenhouse gases changes situated below cloud is minimal. Thus, the optical depth due to the greenhouse species, as well as temperature difference between the emitting surface and the top of the atmosphere, and the region in which the greenhouse species absorbs, are much smaller where cloud is present [*Myhre et al.*, 1998]. The impact of cloud on both $NF_3$ and CFC-115 is similar; because they are both very long-lived species, they are both well mixed in the stratosphere and so the magnitude of downward irradiance due to each species at the tropopause is approximately the same.

### 5.4 Global Warming Potentials

Global warming potential is defined by the expression:


$$\text{GWP} = \frac{\int_0^{\text{TH}} a_\chi [\chi(t)] dt}{\int_0^{\text{TH}} a_r [r(t)] dt} \qquad \text{(E2)}$$

where TH is time horizon; $a_\chi$ is radiative forcing due to a unit increase in atmospheric abundance of the compound (W m$^{-2}$ kg$^{-1}$) and $[\chi(t)]$ is its time-dependent decay in concentration following its instantaneous release at time $t = 0$. The denominator contains the corresponding quantities for $CO_2$ as a reference gas [*Solomon et al.*, 2007]. GWP is the most common metric used by the WMO and IPCC to compare the potency of a greenhouse gas relative to an equivalent emission of $CO_2$ over a set time period. GWP takes into account species lifetime. This means a species with a very high radiative forcing may still have a low GWP if it also has a short atmospheric lifetime. Note that GWP is only one of a range of possible metrics and is not necessarily representative of temperature changes or other climate impacts [*Myhre et al.*, 1998], since it does not account for factors such as changes in emission or the introduction of replacement species. Other criticisms are also discussed in greater detail by *Myhre et al.* [1998].

Table 6 lists the 20, 100 and 500-year GWPs based on cloudy sky adjusted radiative efficiencies of $NF_3$ and CFC-115 compared with the values reported in [*Myhre et al.,* 2013]. The GWPs for $NF_3$ are estimated to be 14600, 19400 and 21400 over 20, 100 and 500 years, respectively. Similarly, the GWPs for CFC-115 are 6120, 8060 and 8630 over 20, 100 and 500 years, respectively. The forcing efficiencies determined in the present study are approximately $10 - 15\%$ higher (shown in Section 5), but the atmospheric lifetimes are significantly shorter than the estimates adopted by the Myhre et al. [2013] (see footnote to Table 6). The effect of the change in radiative efficiency is most obvious for the 20-year GWPs where, because the atmospheric lifetimes of $NF_3$ and CFC-115 are respectively 616 and 492 years, the species do not have time for significant loss to occur. In contrast, the 500 year GWP is more indicative of the impact of the reduced lifetimes.

The trade-off between these competing effects is demonstrated in Table 6 where $NF_3$ and CFC-115 exhibit 20-year GWPs larger by 19% and 15%, respectively, than their IPCC determined values. This difference then decreases for the 100 year GWP (13% and 9%) and the 500 year GWP (3% and -14%). Note that in the case of CFC-115, where the IPCC atmospheric lifetime used to define the GWP is 1700 yr, which is over three times that of our value of 539 yr, the 500 year GWP of 9990 in Myhre et al. [2013] is actually larger than our value of 8630.

## 6 Summary and Conclusions

In this study we have presented updated values for the IR absorption cross-sections and atmospheric lifetimes of $NF_3$ and CFC-115, as well as radiative forcing and radiative efficiencies taking into account stratospheric adjustment and cloudy skies. These values have then been used to obtain values for the 20, 100 and 500 year GWPs of both species. A sensitivity analysis for the forcing calculations relating to tropopause definition and grid resolution has also been provided.

The IR cross sections measured in the present study are generally in good agreement with previous measurements; the resulting radiative forcings and efficiencies are approximately 10% larger than those reported previously, which is likely due to differences in the modeled atmospheric profiles. Atmospheric lifetimes of $(616 \pm 34)$ yr and $(492 \pm 22)$ yr have been determined for $NF_3$ and CFC-115, respectively, using the Whole Atmosphere Community Climate Model (WACCM).




Model diagnostics confirm that CFC-115 is removed faster than $NF_3$ in the atmosphere, except in the altitude region of 50-75 km where photolysis is the dominant removal process. Photolysis is the dominant loss process for $NF_3$ loss (60.0%), while for CFC-115 the reaction with $O(^1D)$ dominates (65.6%). Our overall lifetimes for the two gases are similar (within 10%) to those reported in the recent *SPARC* [2013] assessment, but the contribution from the loss via photolysis is less in the case of

$NF_3$.

Our model results show that omitting the stratospheric adjustment can result in an underestimation of radiative forcing of around $10 - 15\%$ and omitting cloud can result in an overestimation of between $30 - 40\%$ (Tables 4 and 5). These differences are in line previous studies by *Pinnock et al.* [1995] who found an overestimation of $25 - 50\%$ over several RF and IRF calculations for a range of hydro-halocarbons. Our results also show a strong variation of greenhouse gas forcings

with season and latitude, varying by as much as several orders of magnitude. The lifetime results and IR absorption cross sections from the present study indicate global warming potentials (over a 500-year period) for $NF_3$ and CFC-115 of 21400 and 8630, respectively.

**Acknowledgements**

This work was part of the MAPLE project (NE/J008621/1) from the UK Natural Environment Research Council, which also provided a studentship for AT. We thank Prof. William Sturges (University of East Anglia) for supplying a sample of CFC-115 and Dr. Doug Kinnison (NCAR) for helping with WACCM simulations.





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

WMO (2007), Scientific Assessment of Ozone Depletion: 2006, Global Ozone Research and Monitoring Project *Rep. 50*, 571 pp, World Meteorological Organization, Geneva, Switzerland.

WMO (2011), Scientific Assessment of Ozone Depletion: 2010, Global Ozone Research and Monitoring Project *Rep. 52*, 516 pp, World Meteorological Organization, Geneva, Switzerland.





**Tables**

**Table 1**. Metal atom and $O(^1D)$ reactions with $NF_3$ and CFC-115 added to the WACCM chemistry module.

| Reaction | Rate coefficient / $cm^3$ molecule$^{-1}$ s$^{-1}$ | | Source |
|---|---|---|---|
| $O(^1D) + NF_3$ | $(2.0 \pm 0.3) \times 10^{-11}$ | | [*Dillon et al.*, 2011] |
| $Na + NF_3$ | $6.0 \times 10^{-10} \exp(-2240/T) + 10^{-11} \exp(-589/T)$ | $2.3 \times$ | [*Totterdill et al.*, 2014] |
| $K + NF_3$ | $16.0 \times 10^{-10} \exp(-2297/T) + 10^{-11} \exp(-866/T)$ | $1.3 \times$ | [*Totterdill et al.*, 2014] |
| $O(^1D) + CFC-115$ | $(6.5 \pm 0.6) \times 10^{-11} \exp(+30/T)$ | | [*Baasandorj et al.*, 2013] |
| $Na + CFC-115$ | $5.9 \times 10^{-10} \exp(-4257/T) + 10^{-11} \exp(-2093/T)$ | $1.8 \times$ | [*Totterdill et al.*, 2015] |
| $K + CFC-115$ | $1.9 \times 10^{-10} \exp(-1929/T)$ | | [*Totterdill et al.*, 2015] |

**Table 2**. Integrated absorption cross sections of $NF_3$ measured in the present study at 296±2 K (1σ uncertainty ±6% for the major bands), and compared to the previous studies of *Robson et al.* [2006] and *Molina et al.* [1995].

| Band Limits / $cm^{-1}$ | Integrated band cross section / $10^{-18}$ $cm^2$ molec$^{-1}$ $cm^{-1}$ | Ratio to integrated cross section from Robson *et. al* | Ratio to integrated cross section from Molina *et. al* |
|---|---|---|---|
| 600-700 | 0.41 | 1.06 | 1.20 |
| 840-960 | 65.03 | 1.03 | 1.79 |
| 970-1085 | 5.88 | 1.11 | 1.39 |
| 1085-1200 | 0.10 | 0.66 | 0.68 |
| 1330-1440 | 0.08 | 4.35 | 4.76 |
| 1460-1580 | 0.21 | 1.54 | 1.59 |
| 1720-1870 | 0.71 | 0.97 | 1.09 |
| 1890-1970 | 0.65 | 1.01 | 1.06 |
| 600-1970 | 73.50 | 1.04 | 1.72 |



**Table 3**. Integrated absorption cross sections of CFC-115 measured in the present study at 296±2 K (1σ uncertainty ±6% for the major bands), and compared to the previous study of *McDaniel et al.* [1991].

| Band Limits / cm$^{-1}$ | Integrated band cross section / $10^{-17}$ cm$^2$ molec$^{-1}$ cm$^{-1}$ | Ratio to integrated cross section from McDaniel *et. al* |
|---|---|---|
| 946-1020 | 2.546 | 0.91 |
| 1105-1150 | 2.015 | 0.94 |
| 1160-1212 | 1.370 | 0.95 |
| 1212-1265 | 5.381 | 0.94 |
| 1326-1368 | 0.620 | 0.97 |

5   **Table 4.** Instantaneous and stratosphere-adjusted radiative forcings of NF$_3$ and CFC-115 in clear and cloudy sky conditions.

| Molecule | Instantaneous | | Adjusted | |
|---|---|---|---|---|
|  | Clear, $10^{-4}$ Wm$^{-2}$ | Cloudy, $10^{-4}$ Wm$^{-2}$ | Clear, $10^{-4}$ Wm$^{-2}$ | Cloudy, $10^{-4}$ Wm$^{-2}$ |
| NF$_3$ | 3.30 | 2.08 | 3.53 | 2.79 |
| CFC–115 | 27.70 | 18.09 | 29.77 | 19.05 |



**Table 5.** Instantaneous and stratosphere-adjusted radiative efficiencies of $NF_3$ and CFC-115 in clear and cloudy sky conditions.

| Molecule | Instantaneous | | Adjusted | |
| --- | --- | --- | --- | --- |
| | Clear, $W\ m^{-2}\ ppb^{-1}$ | Cloudy, $W\ m^{-2}\ ppb^{-1}$ | Clear, $W\ m^{-2}\ ppb^{-1}$ | Cloudy, $W\ m^{-2}\ ppb^{-1}$ |
| $NF_3$ | 0.35 | 0.22 | 0.40 | 0.25 |
| CFC-115 | 0.32 | 0.20 | 0.35 | 0.21 |

5  **Table 6.** Comparison of the 20, 100 and 500-year global warming potentials for $NF_3$ and CFC-115 from this work the IPCC AR5.

| Molecule | This Work | | | IPCC AR5 | | |
| --- | --- | --- | --- | --- | --- | --- |
| | $GWP_{20}$ | $GWP_{100}$ | $GWP_{500}$ | $GWP_{20}$ | $GWP_{100}$ | $GWP_{500}$ |
| $NF_3$ | 14600 | 19400 | 21400 | 12300[a] | 17200[a] | 20700[a] |
| CFC-115 | 6120 | 8060 | 8630 | 5310[b] | 7370[b] | 9990[b] |

[a] based on an atmospheric lifetime of 740 years.

[b] based on an atmospheric lifetime of 1700 years.


**Figures**

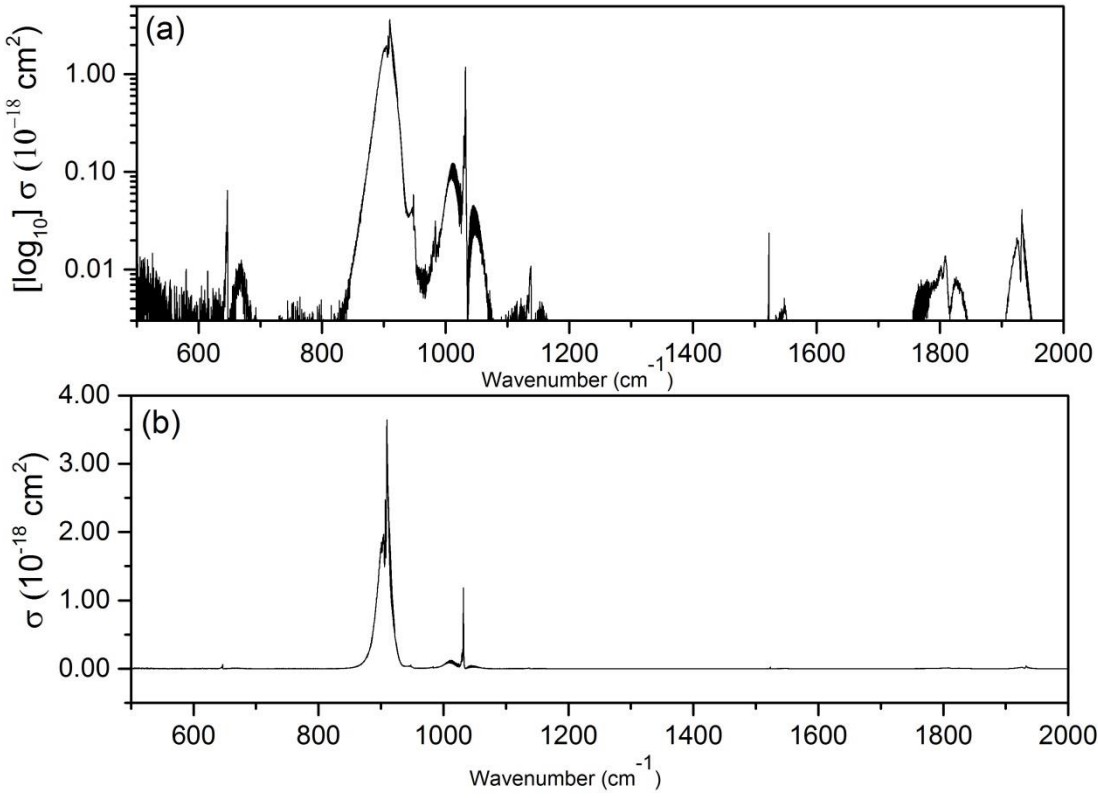

**Figure 1. Infrared absorption spectrum of NF₃ at 295±2 K on a logarithmic (a) and linear (b) scale. The logarithmic plot highlights the minor bands.**



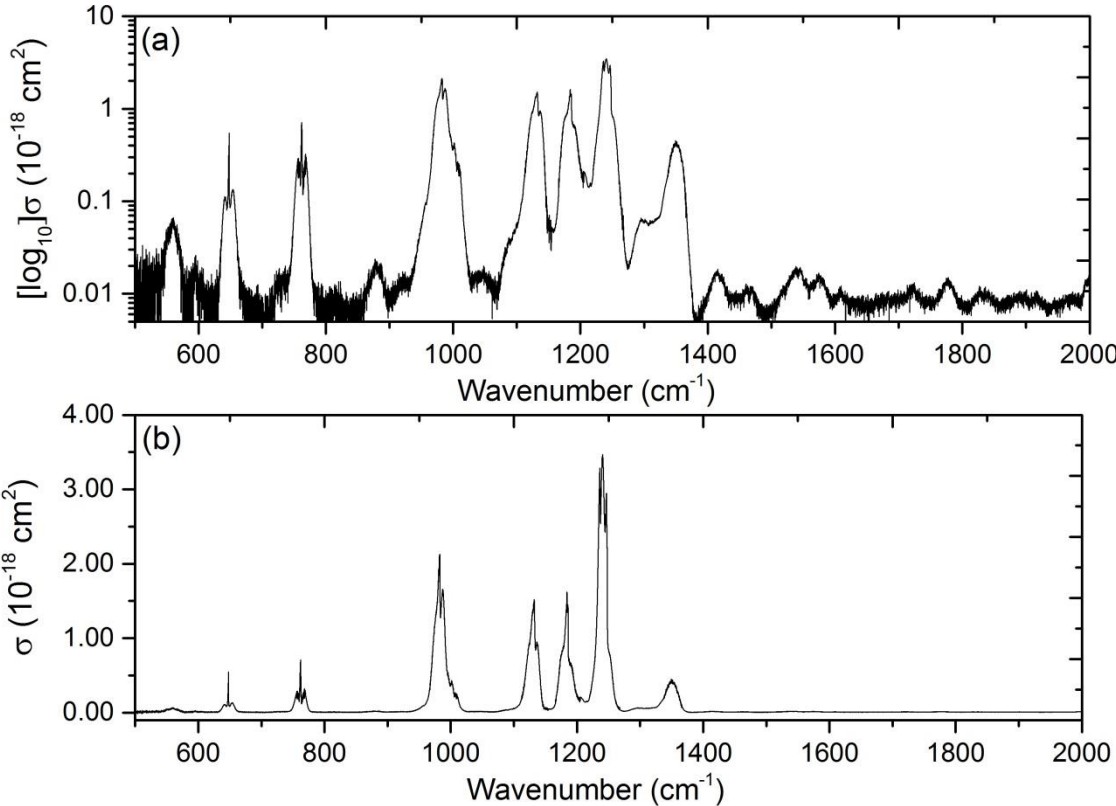

**Figure 2. Infrared absorption spectrum of CFC-115 at 295±2 K on a logarithmic (a) and linear (b) scale. The logarithmic plot highlights the minor bands.**



**Figure 3. (a) Globally averaged NF$_3$ mixing ratio in January of the 13$^{th}$ year of the WACCM simulation, illustrating the profiles of the 5 individual tracers (passive, photolysis, metal reaction and reaction with O($^1$D)). (c) The corresponding zonal mean NF$_3$ mixing ratio (ppt) as a function of altitude. (b) As panel (a) but for CFC-115. (d) As panel (c) but for CFC-115.**







**Figure 4. Atmospheric loss rates of NF$_3$ (left-hand panels) and CFC-115 (right-hand panels) showing the total loss rate, loss rates via photolysis and reaction with O($^1$D), and the percentage contributions to the total loss rates, in the 13$^{th}$ year of the simulation.**





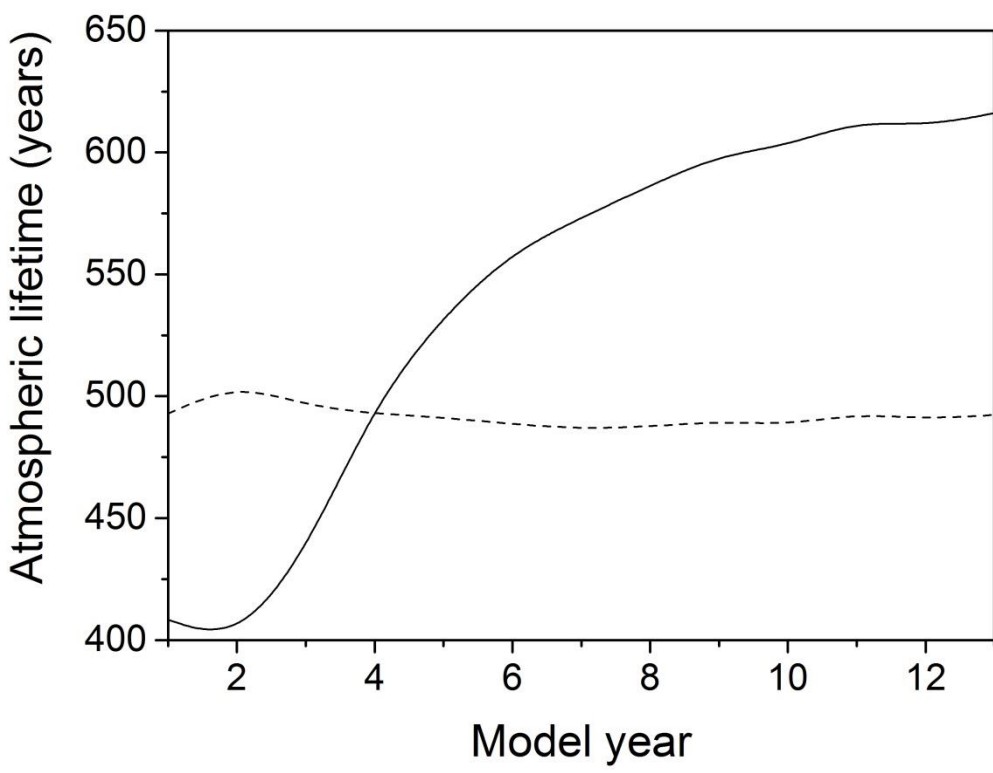

**Figure 5. Annually averaged atmospheric lifetimes for NF₃ (solid line) and CFC-115 (dashed line) as a function of simulation time.**





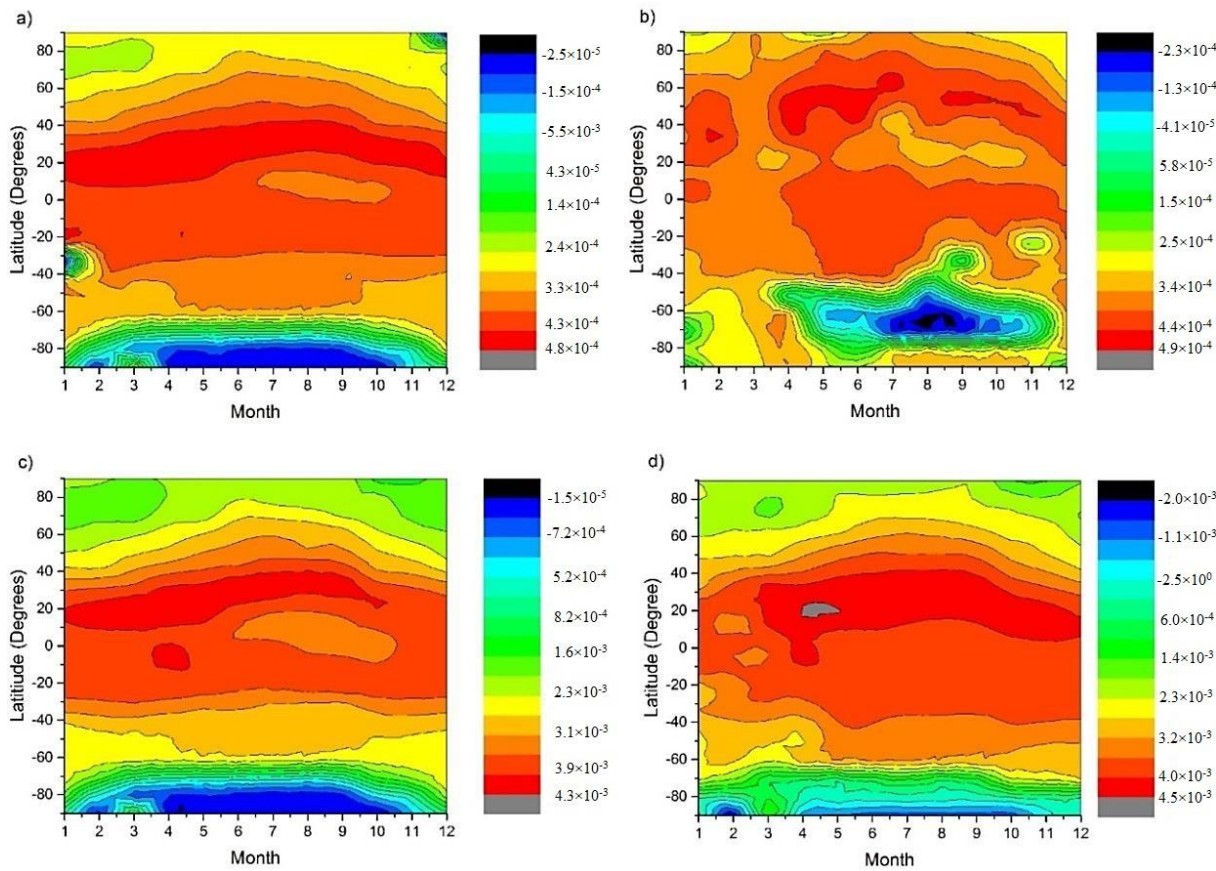

**Figure 6. Contour plots for radiative forcing (Wm$^{-2}$) by latitude and month for (a) instantaneous radiative forcing of NF$_3$, (b) stratospheric adjusted radiative forcing of NF$_3$, (c) instantaneous radiative forcing of CFC-115, and (d) stratospheric adjusted radiative forcing of CFC-115. Note different contour intervals between panels.**




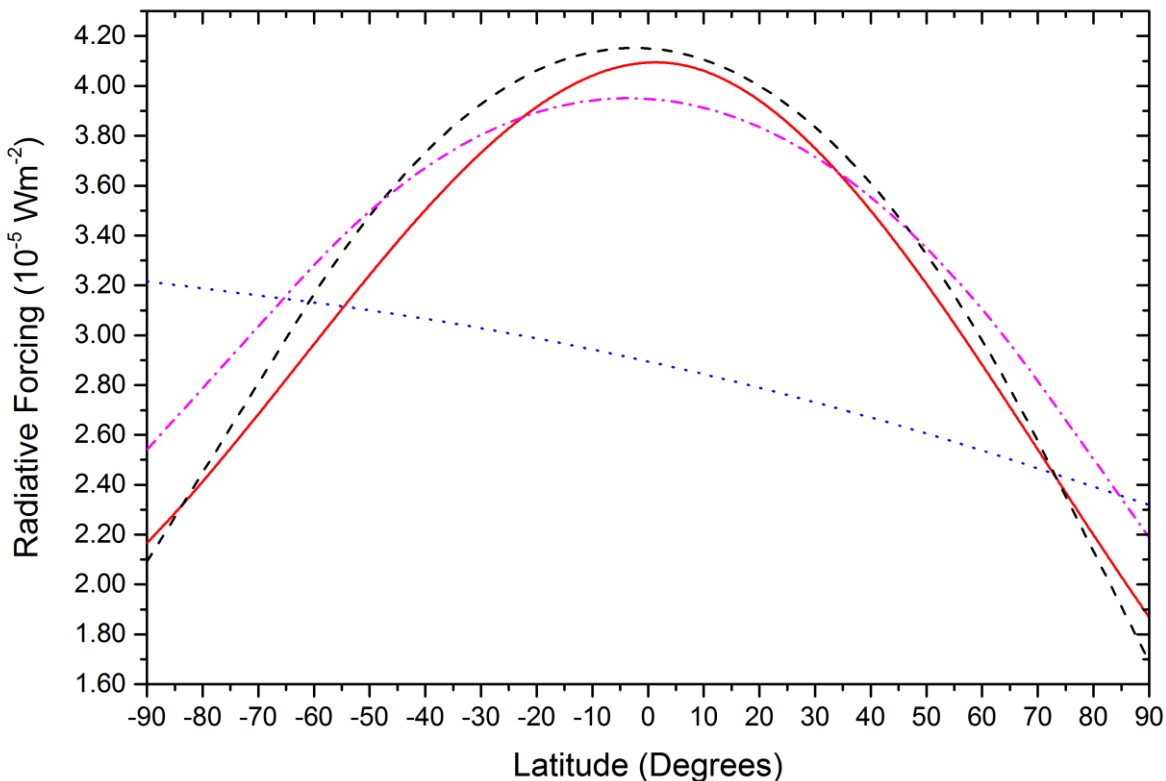

**Figure 7. Variation of instantaneous radiative forcing for NF₃ (Wm⁻²) with tropopause height for four profiles: thermal tropopause (red solid line), average thermal tropopause (black dashed line), temperature minimum tropopause (dash dot magenta line) and average temperature minimum tropopause (blue dotted line).**