# Peer review of "Atmospheric Lifetimes, Infrared Absorption Spectra, Radiative Forcings and Global Warming Potentials of NF3 and CF3CF2Cl (CFC-115)"

_Atmospheric Chemistry and Physics, 2016_

## Author Comment (AC1) · 6 May 2016

We have identified some important errors in our GWP calculations and its comparison to IPCC AR5. The discussion of the NF3 radiative efficiency is also misleading in places. This does not affect the main findings of the paper or our conclusions but it does effect quantitive details.

These errors are summarised below

1. The Abstract and Section 5.3.3. This incorrectly states that NF3 radiative efficiencies are 10% higher than those reported previously. In fact the radiative efficiency for NF3 is 25% higher than that employed in IPCC AR5. The text is also confusing as it is not

[Figure]

made clear by us that the cloudy-sky adjusted radiative efficiency is the most important estimate that is taken forward into the GWP estimate.

2. Our GWP calculations are wrong in Table 6 as IPCC AR4 numbers were inadvertently used for the absolute GWP estimate in the calculation of GWPs. Further, the quoted IPCC AR5 numbers are not in fact from Myhre et al. 2013, they are rather from the earlier IPCC AR4 report (Forster et al. 2007). Our estimates should be updated to the following 15800, 20100 and 22800 for the GWP20, GWP100 and GWP500 of NF3 respectively. And 6080, 7630 and 8080 for the GWP20, GWP100 and GWP500 of CFC-115 respectively. The Table should refer to IPCC AR4 and not IPCC AR5. IPCC AR5 (Myhre et al. 2013) did not present 500 year GWPs.

3. As a result of the updated GWPs and the incorrect IPCC reference some of the discussion in Section 5.4 is quantitatively incorrect and needs to be corrected.

An updated version of Table 6 is attached with both AR4 and AR5 numbers for reference

We apologise to the reviewers and the editor for these mistakes

Please also note the supplement to this comment:
http://www.atmos-chem-phys-discuss.net/acp-2016-231/acp-2016-231-AC1-supplement.pdf

**Supplement:**

**Table 6.** Comparison of the 20, 100 and 500-year global warming potentials for $NF_3$ and CFC-115 from this work the IPCC AR4 [*Forster et al.*, 2007] and AR5 [*Myhre et al.,* 2013].

| Molecule | This Work | | | IPCC AR4 | | | IPCC AR5 | |
|---|---|---|---|---|---|---|---|---|
| | $GWP_{20}$ | $GWP_{100}$ | $GWP_{500}$ | $GWP_{20}$ | $GWP_{100}$ | $GWP_{500}$ | $GWP_{20}$ | $GWP_{100}$ |
| $NF_3$ | 15800 | 20100 | 22800 | 12300[a] | 17200[a] | 20700[a] | 12800[c] | 16100[c] |
| CFC-115 | 6080 | 7630 | 8080 | 5310[b] | 7370[b] | 9990[b] | 5860[d] | 7670[d] |

[a] based on an atmospheric lifetime of 740 years.

[b] based on an atmospheric lifetime of 1700 years.

[c] based on an atmospheric lifetime of 500 years

[d] based on an atmospheric lifetime of 1020 years

Forster, P. M., et al. (2007), Changes in Atmospheric Constituents and in Radiative Forcing, in *Climate Change 2007: The Physical Science Basis. Contribution of Working Group I to the Fourth Assessment Report of the Intergovernmental Panel on Climate Change*, edited by D. Qin, M. Manning, Z. Chen, M. Marquis, K. B. Averyt, M. Tignor and H. L. Miller, Cambridge University Press, Cambridge, United Kingdom and New York, NY, USA.

Myhre, G., D. Shindell, F.-M. Bréon, W. Collins, J. Fuglestvedt, J. Huang, D. Koch, J.-F. Lamarque, D. Lee, B. Mendoza, T. Nakajima, A. Robock, G. Stephens, T. Takemura and H. Zhang, 2013: Anthropogenic and Natural Radiative Forcing. In: Climate Change 2013: The Physical Science Basis. Contribution of Working Group I to the Fifth Assessment Report of the Intergovernmental Panel on Climate Change [Stocker, T.F., D. Qin, G.-K. Plattner, M. Tignor, S.K. Allen, J. Boschung, A. Nauels, Y. Xia, V. Bex and P.M. Midgley (eds.)]. Cambridge University Press, Cambridge, United Kingdom and New York, NY, USA, pp. 659–740, doi:10.1017/ CBO9781107415324.018

---

## Referee Comment (RC1) · Anonymous Referee #1 · 18 May 2016

Interactive comment on "Atmospheric Lifetimes, Infrared Absorption Spectra, Radiative Forcings and Global Warming Potentials of NF3 and CFC-115", by Anna Totterdill et al.

This manuscript discusses updated measurements of infrared absorption spectra of NF 3 and CFC- 115, and calculates the radiative forcing and efficiency of these molecules taking into account the impacts of clouds and stratospheric adjustment. The WACCM model is used to calculate atmospheric lifetimes, and global warming potentials for NF 3 and CFC-115 are also reported. The results of this study are important and will feed into the IPCC and WMO ozone assessment activities. The scope of the paper is clearly relevant for publication in ACP. There are a few issues outlined below which the authors should consider before publication (I have taken into account the authors' corrections

posted on 6 May).

Specific Comments:

p. 2, L19: this should be stated as "based on the previous O( 1 D) reactive yield from Sander et al." – "based on previous photolysis cross sections" was incorrectly stated in the SPARC (2013) report. The CFC-115 cross-sections used in SPARC (2013) were unchanged from Sander et al. (2011).

p. 3, L16: suggest changing "It" to "WACCM". Also, I recommend removing "revised" from this sentence. Stating that these are "revised" estimates of the lifetimes implies that the new values supersede those in SPARC (2013). However, this would require a more in-depth analysis of the updated loss rates than is presented in the paper, i.e., showing that the new loss rates are improved over previously reported vales (see comment below regarding the photolysis rates), and the authors also show that the impacts of the metal reactions (which were not addressed in SPARC (2013)) are negligible.

p. 4, L30-32: In the discussion of the photolysis cross sections, there is no reference of a recent study by Papadimitriou et al. (GRL, 2013, pp. 440-445) who reported updated laboratory measurements of NF 3 photolysis cross sections (these results were also reported in SPARC (2013). The Papadimitriou et al. study showed the importance of the 200-220 nm region for NF 3 photolysis. However, the present paper states that the cross sections used in WACCM are only for 121.6-200 nm based on their previous studies. Neglecting the 200-220 nm region could cause the differences in the NF 3 lifetime and fractional loss contributions discussed on p. 7 and in the Conclusions. The authors should at least briefly mention this point and how the cross sections they used differ from the Papadimitriou et al. study (more specifically than just the general statement at the bottom of p. 7).

I have a similar comment in regards to the CFC-115 cross sections. Please provide a brief statement as to how the values from the previous Totterdill et al. studies used in WACCM differ from current recommended values (Sander et al., 2011 or IUPAC), and

why the 200-230 nm range was not included. SPARC (2013) reported that the 190-230 nm region accounted for 28% of the total CFC-115 loss.

p. 4, L32 - p. 5, L 2: the authors state that they are running the model to steady state, however, it appears that they are using time dependent solar forcing, as opposed to a fixed average solar forcing. Perhaps the time dependent solar forcing does not impact the lifetimes? Please clarify.

p. 5, L26: please provide some justification for the assumption that the temperature dependence contributes negligible uncertainty, eg, is this the case for other molecules?

p. 8, L9-12: should also include that the tropopause can be defined by potential vorticity. How would a tropopause defined by PV impact the radiative forcing calculations? Perhaps the results would be similar to the thermal definitions, but this should be stated.

p. 9, L1: "Figure 7" – this appears to be labeled as Figure 6, and shouldn't this come after the figure currently labeled Figure 7?

p. 9, L3: "The latitudinal variation....as large as a factor of 8". Please be more specific here - is the "factor of 8" the equator-SPole difference?

p. 9, L6-7: "..primarily due to changes in the Planck function." Please provide a little more explanation as to why this is so, e.g., is this due to latitudinal changes in the background ambient temperature?

p. 9, L7-8: regarding the 25% difference in the SH vs. NH. Again, can you provide a little more explanation as to why this is, eg, is this due to the very cold temps of the SH polar vortex? Perhaps this is being implied in the next sentence, but it should be more directly stated.

Technical Corrections:

p. 3, L20: "...some deviation across existing the literature cross-sections" should be corrected – it's unclear what is being said here.

p. 4, L14: "5.96x10-6 Pa" – to help orient the reader, also include the corresponding approximate altitude, ie, ~140 km, judging by Fig. 3.

p. 7, L4: "..tracers to mix vertically into this region." To clarify, I suggest adding "due to the dominance of molecular diffusion" to the end of this sentence.

p. 7, L12: change "CFC-115 uses 1.05 and ..." to "CFC-115 are 1.05 and..."

p. 7, L14: the TTL is ~12-17 km (or there abouts). 20-28 km is much too high.

p. 7, L15: add "mixing" before "ratios"

p. 7, L17: "(see next)" – what does this refer to? The next figure?

p. 8, L4: change "cloud is" to "clouds are"

p. 9, L16-17: add "is" before "obtained"

p. 9, L27: change "gases changes" to "gas changes", and include "a" before "cloud"

p. 9, L29: change "cloud is" to "clouds are"

p. 11, L7: change "cloud" to "clouds"

p. 16-17, captions for Tables 2-3: to clarify, I suggest adding "IR" after "Integrated"

Fig. 4, top 6 panels: it would be more informative to present the losses as mixing ratio/year instead of Tonnes/year.

Fig. 4, bottom 4 panels: to clarify, please add "total" in all figure titles, e.g: "Ratio of total NF3 loss via photolysis(%)"

—————————————————————

---

## Referee Comment (RC2) · Anonymous Referee #2 · 20 May 2016

Authors: Anna Totterdill, Tamás Kovács, Wuhu Feng, Sandip Dhomse, Christopher J. Smith, Juan Carlos Gómez–Martín, Martyn P. Chipperfield, Piers M. Forster and John M. C. Plane

Ms. Title: Atmospheric Lifetimes, Infrared Absorption Spectra, Radiative Forcings and Global Warming Potentials of NF3 and CFC-115

In the present work, the authors report experimentally determined IR cross sections for NF3 and CF3CF2Cl (CFC-115) that they were introduced in two different radiative transfer models to calculate radiative efficiencies and forcings. In this framework, they simulated species' distribution by using a 3-D model (WACCM) and they calculated the atmospheric lifetimes, for both species, by employing a whole atmosphere

chemistry-climate model. Finally, they estimated the global warming potentials (GWP) for NF3 and CFC-115 and they compared the results with WMO, IPCC and SPARC reports, as well as the measured IR cross sections with the previously determined values. Although there are several studies in the literature that estimate NF3 and CFC-115 GWPs, the results and the approach used in this work contribute to better understand the significance of all the parameters that affect the climate impact of those emissions and thus they are worth to be published. However, the present reviewer believes that there are some issues that the authors need to clarify before the current submission would be in a publishable form. Comments and questions are listed in detail below: Minor issues that will help though to improve the quality of the paper are: 1. Although rate constant is commonly used the term is not scientifically correct and should be replaced in the whole text with rate coefficient, since it is not a constant and varies with temperature at least. 2. All the sentences that start with witch and where should include a comma before that, i.e., ,which, throughout the manuscript.

Line by line and general comments that need to be addressed: 1. Pg 1.Title: Please include the formula in the title and use the CFC-115 in parenthesis, i. e., "...of NF3 and CF3CF2Cl (CFC-115)". 2. Pg 3. line 21, Introduction: Please change "trace gas depends in part on" with "trace gas depends, in part, on" 3. Pg 5. line 16, Introduction: "The purpose of this work was to determine new values....": It is not justified that new values are needed, especially since there are many recent studies and panels evaluations that they have taken into account all of them. It is suggested to rephrase that sentence so as to be consisted with what has been actually done in this work, which introduce some new aspects, such as clouds impact in RE, RF and GWP and more complete atmospheric models to calculate NF3 and CFC-115 distributions and atmospheric lifetimes. IR cross-sections has been measured previously and although it is worth to assess the validity of the existing data in the literature, it is not the major issue for the occurred divergences in GWPs. The new in the present work is more the different approach that examines the impact of other processes to RE, RFs, atmospheric lifetimes and GWPs, than the need for obtaining new values. Please modify

accordingly. 4. Pg 6. line 14, Experimental: Is it 40000 or 4000 cm-1. 5. Pg 6. line 17, Experimental: Although the relatively high absorbance for both compounds at the atmospheric window, i.e., 800-1200 cm-1 assist to have high sensitivity (signal to noise ratio) and reliable cross-sections in that range, is that also the case for the lower bands at shorter wavenumbers, with 128 co-added scans at 0.1 cm-1 resolution? How precisely those band strengths were determined? 6. Pg 6. line 21, Experimental: At a selected wavelength or at a selected wavelength range? 7. Pg 6. line 22, Experimental: How the concentration was determined? From the mixing ratios of the manometrically prepared bulbs and the measured pressure? What are the estimated uncertainties? 8. Pg 7. line 1, Experimental: Cross section units are cm2 molecule-1. Please correct. 9. Pg 7. line 4, Experimental: Although it might not be the case here in and no data are depicted to evaluate it, it is not uncommon to observe divergence from Beer-Lambert law at absorbance higher than 0.6. A of 1 corresponds to 90 % loss of the IR light intensity, which is not at the safe end of the Beer-Lambert linearity range. It is important to present cross-section plots in the supplement to demonstrate the validity of the Beer-Lambert in the hole concentration range used. What was the intercept when the experimental data were fitted with a linear function? 10. Pg 7. line 19, Atmospheric Modelling: freeware instead of free running version might be more appropriate. 11. Pg 8. line 1-3 and 21-23, Atmospheric Modelling: Papadimitriou et al. (GRL, 40, 440-445, 2013) demonstrated that Lyman-$\alpha$ is an important loss process for NF3 that account for the $\sim$5 % of its total loss, while NF3 UV spectrum temperature dependence leads to a$\sim$20 % increase of the globally annually averaged lifetime. The authors have neglected both processes and they definitely need to include a reasonable explanation why they have either neglect them or they considered that they will be of minor importance processes. Especially, since they have included in their model processes that have significantly lower contribution to the atmospheric lifetimes, such as mesospheric metals (Na, K) chemistry. The authors need to include the results from the recent studies and to rationalize why they have excluded these two processes or to include them in their model. 12. Pg 9. line 20, Radiative Transfer Modelling: Please

change NF3 to NF3 and CFC115 to CFC-115. 13. Pg 10. line 4-6, Radiative Transfer Modelling: see comment 11. 14. Pg 10. line 23, Results: Infrared Absorption Spectra: Please change "band-integrated cross sections" to "band strengths". 15. Pg 11. line 3, Results: Infrared Absorption Spectra: What are the quoted uncertainties and how they were derived? Are the precisions from the linear fit? 16. Pg 11. line 7-9, Results: Infrared Absorption Spectra: How did the authors estimate the total uncertainties? What are the sources? 17. Pg 13., Results: Atmospheric Lifetimes: A major source for the observed discrepancies, especially between the present results and SPARC report in NF3 results may stem from the Lyman-$\alpha$ and UV temperature dependence ignorance in the present study. (see comment 11) 18. Pg 17. Line 3, Results: Cloudiness: Please change "...efficiencies increase by." with "...efficiencies were increased by...". 19. Pg 18. Line 15, Global Warming Potentials: Please change "...is more indicative." with "...is more representative...". 21. Pg 19. Line 21, Summary and Conclusions: Please change "...in line previous" with "...in line with previous...".

21. Pg 19. Summary and Conclusions: It is necessary the authors to comment on the effect of Lyman-$\alpha$ for both compounds studied in this work and the UV temperature dependence of the NF3 spectrum on their atmospheric lifetimes and either rationalize why they have neglected them or they should include those processes in their models. 22. Figure 2. Remove ticks from mirror axes.

(SEE ALSO ATTACHMENT)

Please also note the supplement to this comment:
http://www.atmos-chem-phys-discuss.net/acp-2016-231/acp-2016-231-RC2-supplement.pdf

---

## Author Response (AR1)

Answers to Reviewers' Comments on Totterdill *et al.*,
Atmospheric Lifetimes, Infrared Absorption Spectra, Radiative Forcings and Global Warming Potentials of $NF_3$ and CFC-115

**Reviewer 1**

Comment 1: this should be stated as "based on the previous $O(^1D)$ reactive yield from Sander et al." – "based on previous photolysis cross sections" was incorrectly stated in the SPARC (2013) report. The CFC-115 cross-sections used in SPARC
10 (2013) were unchanged from Sander et al. (2011).

**Answer 1: Corrections made.**

Comment 2: suggest changing "It" to "WACCM". Also, I recommend removing "revised" from this sentence. Stating that
15 these are "revised" estimates of the lifetimes implies that the new values supersede those in SPARC (2013). However, this would require a more in-depth analysis of the updated loss rates than is presented in the paper, i.e., showing that the new loss rates are improved over previously reported vales (see comment below regarding the photolysis rates), and the authors also show that the impacts of the metal reactions (which were not addressed in SPARC (2013)) are negligible.

**Answer 2: Comment considered and corrections made.**

20 Comment 3: In the discussion of the photolysis cross sections, there is no reference of a recent study by Papadimitriou et al. (GRL, 2013, pp. 440-445) who reported updated laboratory measurements of $NF_3$ photolysis cross sections (these results were also reported in SPARC (2013). The Papadimitriou et al. study showed the importance of the 200-220 nm region for NF 3 photolysis. However, the present paper states that the cross sections used in WACCM are only for 121.6-200 nm based on their previous studies. Neglecting the 200-220 nm region could cause the differences in the $NF_3$ lifetime and fractional
25 loss contributions discussed on p. 7 and in the Conclusions. The authors should at least briefly mention this point and how the cross sections they used differ from the Papadimitriou et al. study (more specifically than just the general statement at the bottom of p. 7).
I have a similar comment in regards to the CFC-115 cross sections. Please provide a brief statement as to how the values from the previous Totterdill et al. studies used in WACCM differ from current recommended values (Sander et al., 2011 or
30 IUPAC), and why the 200-230 nm range was not included. SPARC (2013) reported that the 190-230 nm region accounted for 28% of the total CFC-115 loss.

**Answer 3: Clarification added to the text: "In a recent paper by Papadimitriou et al. a report was given on laboratory based $NF_3$ photolysis cross sections. They concluded that the 200-220 nm wavelength range is an important source of the $NF_3$ photolysis, which has strong temperature dependence, too. In the current study this**
35 **wavelength region was not considered, similarly to the temperature dependence. They also concluded that the Lyman-α photolysis counts for 5% of the total loss and the temperature dependence causes a 20% increase in the global atmospheric lifetime. SPARC (2013) reported that the 190-230 nm region accounts for 28% of the total CFC-115 loss, however in this study we concentrated only to the VUV region up to 190 nm."**

Comment 4: the authors state that they are running the model to steady state, however, it appears that they are using time
40 dependent solar forcing, as opposed to a fixed average solar forcing. Perhaps the time dependent solar forcing does not impact the lifetimes? Please clarify.

**Answer 4: Clarification is added to the text: "Although time dependent solar forcing was used in the simulations, is does not have any noticeable impact on the very long lifetime of these gases."**

Comment 5: please provide some justification for the assumption that the temperature dependence contributes negligible uncertainty, eg, is this the case for other molecules?

**Answer 5: The model was not extended with temperature dependent cross sections. No justification can be given on the accuracy of the negligence of the temperature dependence.**

5 Comment 6: p. 8, L9-12: should also include that the tropopause can be defined by potential vorticity. How would a tropopause defined by PV impact the radiative forcing calculations? Perhaps the results would be similar to the thermal definitions, but this should be stated.

**Answer 6: The PV tropopause does not work near the equator, so the standard definition is the thermal tropopause, used here. We feel that adding more text here would confuse the reader.**

10 Comment 7: "Figure 7" this appears to be labeled as Figure 6, and shouldn't this come after the figure currently labeled Figure 7.

**Answer 7: Corrections made.**

15 Comment 8: The latitudinal variation....as large as a factor of 8". Please be more specific here - is the "factor of 8" the equator-SPole difference?

**Answer 8: Corrections made.**

Comment 10. "..primarily due to changes in the Planck function." Please provide a little more explanation as to why this is
20 so, e.g., is this due to latitudinal changes in the background ambient temperature?

**Answer 10: Yes, it is primarily due to temperature. Text changed to "The variation of radiative forcing and efficiency as a function of latitude is primarily due to changes in the Planck function caused by variation in background temperature"**

Comment 11: p. 9, L7-8: regarding the 25% difference in the SH vs. NH. Again, can you provide a little more explanation as
25 to why this is, eg, is this due to the very cold temps of the SH polar vortex? Perhaps this is being implied in the next sentence, but it should be more directly stated.

**Answer 11: Yes, we now change two sentences to make them more explicit. Text changed to: "Forcings averaged across the Southern Hemisphere were approximately 25% lower than those averaged across the Northern Hemisphere due to its average cooler surface temperature. [Prather and Hsu, 2008].The lowest radiative forcings for**
30 **each month are observed at the South Pole due to its cold surface temperature with the very lowest occurring at the winter Antarctic polar vortex."**

Comment 12: "...some deviation across existing the literature cross-sections" should be corrected – it's unclear what is being said here.

**Answer 12: Text changed to: "The cross sections reported in the literature show some scatter, and few […]"**

35 Comment 13: "$5.96 \times 10^{-6}$ Pa" – to help orient the reader, also include the corresponding approximate altitude, ie, 140 km, judging by Fig. 3.

**Answer 13: Corrections made.**

Comment 14: "..tracers to mix vertically into this region." To clarify, I suggest adding "due to the dominance of molecular diffusion" to the end of this sentence.

5  **Answer 14: Corrections made.**

Comment 15: change "CFC-115 uses 1.05 and ..." to "CFC-115 are 1.05 and..."

**Answer 15: Corrections made.**

Comment 16: "the TTL is 12-17 km (or there abouts). 20-28 km is much too high"

**Answer 16: Corrections made.**

15  Comment 17: "add "mixing" before "ratios""

**Answer 17: Corrections made.**

Comment 18: "(see next)" what does this refer to? The next figure?

**Answer 18: Clarifications made in the text.**

Comment 19: "change "cloud is" to "clouds are""

25  **Answer 19: Corrections made.**

Comment 20: "add "is" before "obtained""

**Answer 20: Corrections made.**

Comment 21: "change "gases changes" to "gas changes", and include "a" before "cloud""

**Answer 21: Corrections made.**

35  Comment 22: "change "cloud is" to "clouds are""

**Answer 22: Corrections made.**

Comment 23: "change "cloud" to "clouds""

40
**Answer 23: Corrections made.**

Comment 24: "captions for Tables 2-3: to clarify, I suggest adding "IR" after "Integrated""

45  **Answer 24: Corrections made.**

Comment 25: "Fig. 4, top 6 panels: it would be more informative to present the losses as mixing ratio/year instead of Tonnes/year."

**Answer 25: We thought that the losses are more informative in terms of Tonnes/year as the pressure changes orders of magnitude in the mesosphere.**

Comment 26: "Fig. 4, bottom 4 panels: to clarify, please add "total" in all figure titles, e.g: "Ratio of total $NF_3$ loss via photolysis(%)""

**Answer 26: Clarification is made in the figure caption. However, it is not necessary to add "total" as the important information is the type of the loss process.**

**Reviewer 2**

Comment 1: Although rate constant is commonly used the term is not scientifically correct and should be replaced in the whole text with rate coefficient, since it is not a constant and varied with temperature at least.

**Answer 1: Corrections made.**

Comment 2: All the sentences that start with witch and where should include a comma before that, i.e., ,which, throughout the manuscript.

**Answer 2: Corrections made.**

Comment 3: Title: Please include the formula in the title and use the CFC-115 in parenthesis.

**Answer 3: Corrections made.**

Comment 4: Introduction: Please change "trace gas depends in part on" with "trace gas depends, in part, on"

**Answer 4: Corrections made.**

Comment 5: Introduction: "The purpose of this work was to determine new values.": It is not justified that new values are needed, especially since there are many recent studies and panels evaluations that they have taken into account all of them. It is suggested to rephrase that sentence so as to be consisted with what has been actually done in this work, which introduce some new aspects, such as clouds impact in RE, RF and GWP and more complete atmospheric models to calculate $NF_3$ and CFC-115 distributions and atmospheric lifetimes. IR cross-sections has been measured previously and although it is worth to assess the validity of the existing data in the literature, it is not the major issue for the occurred divergences in GWPs. The new in the present work is more the different approach that examines the impact of other processes to RE, RFs, atmospheric lifetimes and GWPs, than the need for obtaining new values. Please modify accordingly.

**Answer 5: Corrections made.**

Comment 6: Experimental: Is it 40000 or 4000 cm$^{-1}$?

**Answer 6: 4000 cm$^{-1}$.**

Comment 7: Experimental: Although the relatively high absorbance for both compounds at the atmospheric window, i.e., 800-1200 cm-1 assist to have high sensitivity (signal to noise ratio) and reliable cross-sections in that range, is that also the case for the lower bands at shorter wavenumbers, with 128 co-added scans at 0.1 cm-1 resolution? How precisely those band strengths were determined?

**Answer 7: Gas mixtures of different strength where prepared to obtain absorbances with high signal to noise ratio, while still being within the range of validity of the Beer-Lambert law (we mention about the gas mixtures in the text, but we do not say why, so we have updated the text with this explanation).**

Comment 8: Experimental: At a selected wavelength or at a selected wavelength range?

**Answer 8: Wavelength range.**

Comment 9: Experimental: How the concentration was determined? From the mixing ratios of the manometrically prepared bulbs and the measured pressure? What are the estimated uncertainties?

**Answer 9: Text changed: "concentrations were determined from the mixing ratios calculated from pressures measured with a capacitance manometer […]". Uncertainties in concentrations are smaller than 1% are reported in the results section.**

Comment 10: Experimental: Cross section units are $cm^2$ molecule$^{-1}$.

**Answer 11: Molecular cross section units are $cm^2$.**

Comment 11 It is important to present cross-section plots in the supplement to demonstrate the validity of the Beer-Lambert in the hole concentration range used. What was the intercept when the experimental data were fitted with a linear function?

**Answer 11: The absorbance vs. concentration plots were linear and absorbances were always below 0.5.**

Comment 12: Atmospheric Modelling: freeware instead of free running version might be more appropriate

**Answer 12: No, usually "free running" terminology is used.**

Comment 13: Atmospheric Modelling: Papadimitriou et al. (GRL, 40, 440-445, 2013) demonstrated that Lyman-α is an important loss process for $NF_3$ that account for the 5 % of its total loss, while $NF_3$ UV spectrum temperature dependence leads to a 20 % increase of the globally annually averaged lifetime. The authors have neglected both processes and they definitely need to include a reasonable explanation why they have either neglect them or they considered that they will be of minor importance processes. Especially, since they have included in their model processes that have significantly lower contribution to the atmospheric lifetimes, such as mesospheric metals (Na, K) chemistry. The authors need to include the results from the recent studies and to rationalize why they have excluded these two processes or to include them in their model.

**Answer 13: Clarifications made. See "Answer 3" for Reviewer 1.**

Comment 14: Radiative Transfer Modelling: Please change NF3 to $NF_3$ and CFC115 to CFC-115.

**Answer 14: Corrections made.**

Comment 15: Results: Infrared Absorption Spectra: Please change "band-integrated cross sections" to "band strengths".

**Answer 15: Corrections made.**

Comment 16: Results: Infrared Absorption Spectra: What are the quoted uncertainties and how they were derived? Are the precisions from the linear fit? Pg 11. line 7-9, Results: Infrared Absorption Spectra: How did the authors estimate the total uncertainties? What are the sources?

**Answer 16: Sources of uncertainty are stated in Section 5.1. They include uncertainty in the concentration, uncertainty in the linear fit and spectral noise.**

**Admittedly, the statement about the uncertainty in the "scaling" is rather confusing, so the sentence was changed from "The error in scaling the cross sections for NF$_3$ and CFC-115 were estimated to be ±5 and 6%, respectively (error in pressure dependence)" to: "The average standard errors of the slopes obtained from the regression of absorbance versus concentration for NF$_3$ and CFC-115 at selected wavelengths (equation E1) were ±5 and 6%, respectively."**

Comment 17: Results: Atmospheric Lifetimes: A major source for the observed discrepancies, especially between the present results and SPARC report in NF$_3$ results may stem from the Lyman-alpha and UV temperature dependence ignorance in the present study.

**Answer 17: Clarification is given in the text. For more details see "Answer 3" for Reviewer 1.**

Comment 18: Results: Cloudiness: Please change "efficiencies increase by." with "efficiencies were increased by".

**Answer 18: Corrections made.**

Comment 19: Global Warming Potentials: Please change "is more indicative." With "is more representative".

**Answer 19: Corrections made.**

Comment 20: Summary and Conclusions: Please change "in line previous" with "in line with previous".

**Answer 20: Corrections made.**

Comment 21: Summary and Conclusions: It is necessary the authors to comment on the effect of Lyman-α for both compounds studied in this work and the UV temperature dependence of the NF$_3$ spectrum on their atmospheric lifetimes and either rationalize why they have neglected them or they should include those processes in their models.

**Answer 21: Clarification is given in the text. For more details see "Answer 3" for Reviewer 1.**

Comment 22: Remove ticks from mirror axes.

**Answer 22: Corrections made.**

[revised manuscript text omitted]

---

## Author Response (AR2)

Response to the Editor's Comments on Totterdill *et al.*, Atmospheric Lifetimes, Infrared Absorption Spectra, Radiative Forcings and Global Warming Potentials of $NF_3$ and CFC-115

**Editor**

*Comment 1: The lifetimes and subsequent metrics for the title molecules calculated in this work have neglected to include their atmospheric loss due to UV photolysis in the 200 to 230 nm region as stated in the revised submission: "Note that in our modeling we neglected the photolysis caused by 200-230 photons and also the temperature dependence of the UV photolysis. This causes 20% error on the global lifetime values.". It is not sufficient to simply state that the UV photolysis in this region was neglected when the SPARC (2013) report, which is cited in this paper, clearly demonstrated its significance with regard to determining atmospheric lifetimes. Therefore, the lifetimes and dependent radiative parameters reported in this work are, most likely, over estimated and the comparison with previously reported lifetimes and loss processes in Section 5.2 Atmospheric Lifetimes not valid. Stating in the conclusions that the lifetimes are in error by 20% is not sufficient (also it is not clear whether a correction was applied to the reported values, or not). The authors have used a more "sophisticated" model than applied in previous studies, but the impact of these calculations is significantly diminished if the up to date recommendations for atmospheric loss processes are not included. Therefore, I recommend the authors revise the paper to include the neglected photolysis processes prior to consideration for publication.*

**Response**: thank you for making this important point. In the case of CFC-115, the long wavelength (200-230 nm) absorption cross sections is already included, based on Papadimitriou *et al.* (2013). This means that the data on CFC-115 does not change. However, for $NF_3$ the cross sections above 200 nm were neglected. Therefore, the WACCM run was repeated with the extended cross sections. The revised atmospheric lifetime of $NF_3$ is $(509 \pm 21)$ years, which is a 15% reduction on the lifetime if the long wavelength photolysis is not included. Photolysis contributes 67.7% to the $NF_3$ removed lifetime, and reaction with $O(^1D)$ contributes 32.3%. The increased impact of photolysis also changes the atmospheric mixing ratios, so that Figures 3 and 4 have changed. All the latest changes, including the revised Radiative Forcing and Global Warming Potentials, are highlighted with green shaded text in the manuscript.

---

## Author Response (AR3)

Response#2 to the Editor's Comments on Totterdill *et al.*, Atmospheric Lifetimes, Infrared Absorption Spectra, Radiative Forcings and Global Warming Potentials of NF$_3$ and CFC-115

**Editor**

*Comment 1: The comparison of the integrated absorption cross sections of NF$_3$ given in Table 2 show a large discrepancy with previous literature values over the 1330-1440 cm$^{-1}$ region (a very weak band). This band is approximately equal in strength to the band in the region 1085-1200 cm$^{-1}$. However, the log plot shown in Figure 1 does not show the absorption in the 1330-1440 cm$^{-1}$ region. I assume that this is a simple baseline or offset problem with the experimental data. I recommend revising the data shown in Figure 1 (either correcting the baseline or expanding the range of the y-axis) to show this weak band (this is the point of including a log plot). Also, digitized spectra for NF$_3$ and CFC-115 should be included as supplementary material for future reference. I trust that this can be done without further review.*

**Response**: thank you for making this important point.

Older versions of the spectra of NF$_3$ and CFC-115 had been plotted by mistake. The latest versions are now plotted, and the corresponding data is uploaded as supplementary material. The axis of Figure 1, panel a, has been expanded as requested to show the region highlighted by the Editor. The labels of the y axes in Fig. 1 and 2 were wrong and have been changed accordingly.

The NF$_3$ band in the 1330-1440 cm$^{-1}$ range is not as strong as the one in the 1085-1200 cm$^{-1}$ range. The problem does not seem to be related to the baseline correction. Pressure and resolution effects can also be discarded since our spectrum has been measured at the same pressure and with a better resolution than in Robson *et al.*, 2006.